# Positive Effects of a Lecithin-Based Delivery Form of *Boswellia serrata* Extract in Acute Diarrhea of Adult Subjects

**DOI:** 10.3390/nu14091858

**Published:** 2022-04-29

**Authors:** Attilio Giacosa, Antonella Riva, Giovanna Petrangolini, Pietro Allegrini, Teresa Fazia, Luisa Bernardinelli, Gabriella Peroni, Mariangela Rondanelli

**Affiliations:** 1CDI (Centro Diagnostico Italiano), 20147 Milan, Italy; attilio.giacosa@gmail.com; 2R&D Department, Indena SpA, 20139 Milan, Italy; antonella.riva@indena.com (A.R.); giovanna.petrangolini@indena.com (G.P.); pietro.allegrini@indena.com (P.A.); 3Department of Brain and Behavioral Science, University of Pavia, 27100 Pavia, Italy; teresa.fazia01@ateneopv.it (T.F.); luisa.bernardinelli@unipv.it (L.B.); 4Endocrinology and Nutrition Unit, Azienda di Servizi alla Persona ‘‘Istituto Santa Margherita’’, University of Pavia, 27100 Pavia, Italy; 5IRCCS Mondino Foundation, 27100 Pavia, Italy; mariangela.rondanelli@unipv.it; 6Department of Public Health, Experimental and Forensic Medicine, Unit of Human and Clinical Nutrition, University of Pavia, 27100 Pavia, Italy

**Keywords:** acute diarrhea, adult subjects, *Boswellia serrata*, Phytosome, randomized study

## Abstract

Acute diarrhea is a frequent problem worldwide, mostly due to gastrointestinal infections or food poisoning. *Boswellia serrata* could be active in the treatment of acute diarrhea due to its anti-inflammatory, antispasmodic, and antimicrobial activity. In this randomized, double-blind, placebo-controlled clinical study, 49 adults with acute diarrhea were randomly allocated to receive 250 mg of a lecithin-based delivery form of *Boswellia serrata* (CASP) or placebo for 5 days. The time it took to become healthy with stoppage of diarrhea (primary end point) was significantly shorter in the intervention group (3.08 vs. 4.44 days: *p*-value < 0.0001). The probability of subjects treated with CASP to recover sooner was equal to 80.2%. A significantly lower number of stools was observed in the CASP group over time (β = −0.17, *p*-value < 0.0001). A significant difference was observed between the two groups for abdominal pain, nausea, and GAE (global assessment of efficacy). In conclusion, the lecithin-based delivery form of *Boswellia serrata* extract could be a useful addition to the treatment of acute diarrhea in adults. CASP is safe and reduces the time it takes to become healthy, the frequency of stools, the abdominal pain and nausea of subjects with acute diarrhea. Further studies are needed to confirm these promising results.

## 1. Introduction

Acute diarrhea is usually defined as the sudden onset of 3 or more loose stools in a 24-h period, lasting less than 14 days [1]. These cases are characterized by an increase in the volume, frequency, and water content of stools. Abdominal pain, fever, nausea, vomiting and fatigue can be associated symptoms. The most common causes of acute diarrhea are gastrointestinal infections or food poisoning. Viruses represent the most frequent etiologic factor. Every year, 2.5 million people die from acute diarrheal disease worldwide [2]. Most of the cases are observed in the developing world, due to infectious diseases being associated with contaminated food or water [3,4,5]. In the United States, 48 million people suffer every year from acute diarrhea due to acute food poisoning [3,4,6].

Most individuals with acute diarrhea do not need laboratory evaluation and stool cultures. Diagnostic workup should be considered only in patients with dehydration, high fever, blood in stools, immunosuppression syndrome, and when hospital infection is suspected or when symptoms are observed after antibiotic treatment [7,8,9,10,11].

The pathogenesis of acute diarrhea may be of osmotic or secretory origin. Osmotic diarrhea is observed when a nutritional component is not absorbed in the proper way, and water is retained in the gut lumen: a typical example occurs with lactose malabsorption. Malabsorption and osmotic diarrhea may be a consequence of gut viral (rotavirus) or bacterial (*Shigella* spp.’s toxin) loads. Secretory diarrhea occurs when a water secretion into the gut lumen is induced by the presence of specific bacterial toxins [12].

Antibiotics have to be used only when a specific infection has been proven as shigellosis, campylobacteriosis, *Clostridium difficile*, traveler’s diarrhea, and protozoal infections [7,12].

*Boswellia serrata*, also known as Indian frankincense, is a medium to large-sized deciduous and balsamiferous tree: its extract has been used for centuries in Asian and African folk medicine. *Boswellia serrata* extract (BSE) is believed to treat chronic inflammatory diseases as well as various other illnesses. In particular, it has been considered as a potential treatment for inflammatory bowel diseases, including Crohn’s disease and ulcerative colitis. Additional studies showed that BSE is potentially active in the treatment of diarrhea due to its anti-inflammatory, antispasmodic, and antimicrobial activity [13,14]. Based on this scientific evidence, the aim of this paper is to evaluate for the first time the effect of a formulation of Boswellia extract (CASP, Casperome^®^) in the relief of subjects with acute diarrhea.

## 2. Materials and Methods

### 2.1. Study Design

This is a randomized, double-blind, placebo-controlled clinical study. Individuals were randomly allocated to receive CASP supplement or placebo. The study has been performed at the Department of Public Health of the University of Pavia, Italy. The objective of the study was to evaluate the efficacy and safety of CASP in subjects with acute diarrhea. The study was conducted in accordance with the Declaration of Helsinki and the ICH Guidelines for Good Clinical Practice, following approval from the Local Independent Ethics Committee (Ethic code number: 4806/11012022) (approval date is 01 February 2022) and registered at Clinicaltrials.gov (NCT05345028). Written informed consent was obtained from each participant. The study was conducted from 3 March 2021 to 7 September 2021.

### 2.2. Population

Male and female subjects, with a diagnosis of acute diarrhea, were potential candidates for the study. The inclusion criteria were as follows: age 18–70 years; presence of acute diarrhea of mild to moderate severity, that is of clinical conditions that do not require hospitalization or immediate diagnostic laboratory tests [7]. Acute diarrhea was defined as the occurrence of at least three liquid or soft stools per day for no more than 5 days [7].

Exclusion criteria were: patients with long term diarrhea, a concomitant illness that could cause diarrhea, iatrogenic diarrhea, symptoms of functional gut disease or severe inter-current diseases, severe diarrhea and dehydration requiring hospitalization and intravenous treatment, or subjects with high temperature (>38.5 °C). Moreover, subjects treated by other anti-diarrheal and antibiotic drugs in the previous 24 h, as well as severely malnourished subjects and pregnant women, were excluded from the study. Treatment was discontinued in individuals who started taking other drugs during the trial.

### 2.3. Treatment and Concomitant Medications

Subjects included in the study received boswellia Phytosome^®^ (Casperome^®^, Indena S.p.A, Milano, Italy, CASP) standardized to contain ≥ 25% of triterpenoid acids 250 mg or placebo, bid for 5 days. The choice of dose was based on previous clinical experience with CASP [13,14]. Tablets with no active ingredient were used as placebo and were identical to CASP ones in terms of size, shape, color, odor, and taste. CASP and placebo film-coated tablets had similar composition in terms of inactive components. Supplementation compliance was calculated as the ratio between the administered supplement (as determined by returned tablets) and the expected intake during the actual treatment period for each subject. All of the subjects (cases supplemented with CASP and placebo controls) were treated with ORS (Oral Rehydration Solutions), according to World Health Organization indications [15].

Moreover, the BRAT diet (bananas, rice, applesauce, and toast) and the avoidance of fresh dairy products as well as of fiber-rich products was recommended in all individuals. Treatment with anti-motility, probiotic, adsorbent, spasmolytic, antibiotic, or sulphonamide agents was not permitted during the trial.

### 2.4. Clinical Evaluation

Study visits were scheduled at baseline (day 0) and at day 6. The data collection on day 0 included date of onset of diarrhea, previous treatment, number and consistency of stools, body temperature, signs of dehydration and any other data by clinical examination.

During the second visit (day 6), the following data were collected and recorded in the study record forms: (1): date of stoppage of diarrhea in case of inter-current recovery, daily record of frequency and consistency of stools, tolerance and acceptability of treatment; (2) abdominal distension, nausea, and vomiting were evaluated according to a 4-point Likert scale ranging from absence of symptoms (0 points) to severe symptoms (3 points) [15]; (3) abdominal pain and intensity: the score was defined using a validated visual analog scale score for pain (0: ‘no pain’, 10: ‘most severe pain’) [16,17]; (4) Global Assessment of Efficacy (GAE): a 4-point scale was defined by each subject (1: ‘ineffective’, 2: ‘moderately effective slight improvement of complaints’, 3: ‘effective marked improvement in symptoms’, 4: ‘very effective—as good as no symptoms’) [18].

Participants were instructed to promptly report of any unwanted discomfort during the 5-day supplementation, intended as an adverse event. Adverse events were likewise checked at the final visit.

Subjects with worsening diarrhea or worsening clinical conditions (dehydration, persistent fever, bloody diarrhea) dropped out of the study and underwent laboratory evaluation, stool culture, specific pharmacological treatment, and possible hospitalization.

### 2.5. Supplement Description

For the clinical study, a sunflower lecithin-based formulation of *Boswellia serrata* extract (Boswellia Phytosome^®^, Casperome, Indena S.p.A, Milano, Italy, CASP) standardized to contain ≥25% of triterpenoid acids was prepared by Indena S.p.A., as oblong-shaped, film-coated tablets of 250 mg. CASP was formulated as film-coated tablets containing food-grade ingredients; and placebo film-coated tablets was formulated in order to have the same shape and color of CASP tablets. All of the film-coated tablets were characterized for appearance, average mass, uniformity of mass, HPLC content of boswellic acids (only for CASP tablets), disintegration time and microbiological quality.

### 2.6. Study Endpoints

The primary endpoint was time to become healthy (day of stoppage of diarrhea). The end of diarrhea was defined as: the occurrence of the first formed stool or a period of 12 h without any liquid or soft stools. Secondary endpoints were self-assessed daily number of stools, percentage of subjects without diarrhea within 5 days of treatment, concomitant symptoms at the end of study (abdominal pain, abdominal distension, nausea, vomiting) and Global Assessment of Efficacy (GAE).

### 2.7. Statistical Analysis

The sample size was estimated for the primary endpoint time of becoming healthy: the stoppage of diarrhea. Considering a Hazard Ratio (HR) of 3, that implies a 75% of probability of healing earlier in the supplemented than in the placebo group. With a censoring rate of 0.10, a follow-up of 5 days and a median survival time in the placebo group of 4 days, with 80% statistical power at the 5% significance level, a total sample size of 40 subjects (20 per groups) is required. Differences between baseline variables in the two groups were investigated using t-tests for continuous variables and chi-squared or Fisher exact test for categorical ones.

As for the primary endpoint time of becoming healthy, the stoppage of diarrhea, the survival distribution of the duration of diarrhea was estimated for each group using Kaplan-Meier, and the curves were compared using the log-rank test to test the null hypothesis of no difference in the two curves between the two groups. A Cox proportional hazards model for time to recovery from diarrhea adjusted for age and sex was also fitted.

To evaluate statistically significant changes over time in the number of stools in the two groups, a generalized linear mixed-effects model for longitudinal data was fitted. In the model, we specified as fixed effect time, group and the interaction between time and group, and the subject as random intercept to allow for intra-subject correlation produced by the repeated measurements carried out on the same subjects. The model was adjusted for age and sex. The coefficient of the interaction between time and group measures the difference in slopes between the two groups. It also estimates the effect of the treatment on the reduction in the number of stools, thus indicating how much more the treatment group is improving over time with respect to the investigated endpoint and compared to the placebo group over the same time period.

The Fisher’s exact test was used to compare proportions of GAE, abdominal pain, abdominal distension, nausea, and vomiting in the two groups.

All of the analysis was performed using R 3.5.1 statistical software [19]. Descriptive statistics are reported as Mean ± Standard Deviation (SD) and frequency distribution.

## 3. Results

A total of 73 subjects with acute diarrhea were assessed for eligibility. Moreover, 24 subjects were excluded: 15 subjects did not meet the inclusion criteria and 9 declined to participate (Figure 1).

A total of 49 individuals with acute diarrhea were analyzed, 27 females and 22 males, with mean (±SD) age of 50 (±15). Furthermore, 25 subjects were randomly assigned to the placebo group, while 24 to the supplemented group. A total of 2 subjects in the placebo group (8%) dropped out, after three and four days of intervention, respectively, due to worsening symptoms. In Table 1, the descriptive statistics of the baseline variables are reported separately for the two groups, along with the *p*-values for group comparisons. No statistically significant differences were observed between the two groups at baseline.

For the primary endpoint time of becoming healthy (stoppage of diarrhea), the Kaplan-Maier curve for the two groups, showing the cumulative event probability against time, is shown in Figure 2. The log-rank test of the 2 curves showed the 2 groups to be significantly different (*p* < 0.0001), with the mean time of becoming healthy (average resolution time) being 3.08 days and with a median time (at which 50% of cases are resolved) of 3 days for the supplemented group. For the placebo group, the mean time of reaching health was of 4.44 days with a median of 5 days (Figure 2, Table 2).

The Cox proportional hazards model showed a statistically significant HR (adjusted for sex and age) of 4.05 (95%CI, 2.07–7.96), with a *p*-value < 0.0001, representing the odds that a treated subject will resolve diarrhea sooner than a person in the placebo group, corresponding to the probability of subjects treated with CASP to recover sooner equal to 80.2%.

The percentage of subjects’ diarrhea-free within 5 days of treatment was 95.8% in the treated group and 76.0% in the placebo group. In Figure 3, the mean of self-assessed number of stools over time in the two groups is reported. A statistically significant difference between the 2 groups (β = −0.17, *p*-value < 0.0001) was observed.

As reported in Table 3, at the end of the study, a significant difference was observed between the two groups for abdominal pain, nausea, and GAE. While no difference between the two groups was observed for abdominal distension and vomiting.

## 4. Discussion

This study demonstrates that the supplementation with the sunflower lecithin-based formulation of CASP is associated with a significant reduction in the time to be rid of acute diarrhea as compared to placebo. The reported mean time to become healthy permits an ameliorated average resolution time of acute diarrhea after supplementation in respect to the placebo, where more than one day of relief is important for this condition. Significantly, at the end of the study, the percentage of subjects free from diarrhea within 5 days of supplementation was 95.8% in the treated group and 76% in the placebo group. Two individuals in the placebo group (8%) interrupted the supplementation after 3 and 4 days of intervention due to worsening clinical conditions, with evidence of dehydration, and were hospitalized, while none of the subjects in the intervention group interrupted the study. The probability of subjects treated with CASP to recover sooner as compared to controls treated with placebo was equal to 80.2%.

The time it took to become healthy, that is the duration of acute diarrhea, was the primary endpoint of the study, and the data show that this goal has been significantly achieved, thus proving that the supplementation with CASP is a new potential approach to relief this particularly uncomfortable clinical problem.

Our study showed a significant reduction in the number of stools for the supplemented group. The difference started to be statistically significant after two days of intervention and persisted until the end of the study, that is, at the fifth day of supplementation

The usual treatment of acute diarrhea is based on medications such as loperamide and racecadotril [7]. Loperamide is an anti-motility drug that can reduce the duration of diarrhea by as much as one day and may increase the treatment efficacy when administered in combination with antibiotics for traveler’s diarrhea [20,21]. Racecadotril is an anti-secretory agent as effective as loperamide in the treatment of acute diarrhea, with the advantage to be more tolerable and safe [22]. Future studies are needed to compare the effect of CASP with loperamide and racecadotril alone, or in combination with CASP. Antibiotics are routinely considered only in subjects older than 65 years or in immunocompromised, severely ill, or septic subjects [7].

Probiotics are frequently used in subjects with acute diarrhea because they are thought to have an anti-pathogen activity, to stimulate the mucosal immune responses and to reduce the permeability and inflammation of gut mucosa [23,24,25]. There is high-quality evidence that probiotics are effective for acute infectious diarrhea in adults. [7,26,27].

A recent review published by S. Collinson et. Al suggested that it could be uncertain if probiotics actually reduce the duration of acute infectious diarrhea [28]. Nevertheless, the possibility of using probiotics for the restoration of the intestinal flora, in combination with botanical ingredients (such as *Boswellia serrata* extracts) able to target the inflammatory aspect and contractile dysfunctions of the intestine, opens new scenarios in the symptomatic approach to gastrointestinal dysfunctions.

The potential explanation of the positive results obtained in this study with CASP supplementation in subjects with acute diarrhea could be due to the anti-inflammatory, and especially antimicrobial, activity of boswellia extracts [29,30,31]. The bioactive compounds of *Boswellia serrata* extract are triterpenes (30–60%) such as α- and β-boswellic acid and lupeolic acid, essential oils (5–10%), and polysaccharides. The most active phytochemicals of BSE are 11-keto-β-boswellic acid (KBA) and acetyl-11-keto-β-boswellic acid (AKBA) [32]. Multiple anti-inflammatory effects of *Boswellia serrata* extracts have been shown such as the inhibition of 5-lipoxygenase and various effects on the immune system. Among the immunological effects of BSE, the reduction in several cytokines and in particular of TNF-α and interleukins, the decrease in the complement system and of the leukocyte elastase activity, the decrease in ROS formation and of P-selectin-mediated recruitment of inflammatory cells have been demonstrated [33,34,35].

Antimicrobial activity of BSE against *Staphylococcus aureus**, Escherichia coli*, [36] and *Streptococcus mutans* and its antifungal activity against *Candida albicans* [36] has been confirmed. *coli*, *Klebsiella pneumonia*, *Enterobacter aerogenes*, *Pseudomonas aeruginosa* and *Proteus vulgaris*) bacterial strains. [37] The degree of inhibitory activity of the BSE against various bacteria was: *E. coli* > *S. aureus* > *Bacillus subtilis* > *Salmonella typhi* > *Klebsiella pneumoniae* > *Streptococcus pneumoniae* > *Enterobacter aerogenes* > *Proteus vulgaris* [37].

The demonstrated anti-microbial effect of BSE could target the pathogenic bacteria involved in the inflammatory aspect of the intestine [32]. S.M, Isnail et al. showed a significant anti-bacterial activity of *Boswellia serrata* extract [37]. Their in vitro screening test showed a significant bacteriostatic effect of BSE with gram-positive (*Bacillus subtilis*, *S. aureus* and *Streptococcus pneumonia*) and gram negative (Eetiology of various cases of acute diarrhea. Moreover, BSE could be useful to favor the management of subjects with acute diarrhea because it may preserve the intestinal epithelial barrier from oxidative and inflammatory damage [38]. BSE and its selected component acetyl-11-keto-β-boswellic acid reduce the amount of ROS after H_2_O_2_ exposure. This in vitro model of gut inflammation let us hypothesize that CASP may favor the integrity of the structure and function of the intestinal barrier. The BSE protection of intestinal epithelial barrier from inflammatory damage supports its efficacious supplementation in subjects with acute diarrhea similarly to what already demonstrated with CASP in subjects with inflammatory bowel disease [13,14,39].

CASP may also be responsible for favorable effects on intestinal motility of subjects with acute diarrhea. In an animal model, BSE inhibited gastrointestinal transit time in croton oil as well as castor oil-induced diarrhea with a mechanism involving L-type Ca(2+) channels [40].

At the end of our study, the supplementation with CASP was followed by a significant reduction in abdominal pain and nausea as compared to placebo, while no difference was observed on abdominal distension and vomiting. The effect on abdominal pain may be due to the anti-inflammatory effect of *Boswellia serrata* extracts, while the effect on nausea could be favored by the effects of BSE on gut motility [38,40].

The positive outcome of this botanical management of acute diarrhea is confirmed by the participants’ global assessment of efficacy (GAE) that showed a significant difference when cases and controls were compared. At the end of the study, 87.5% of the CASP group were classified in grade 3 (‘effective- marked improvement in symptoms’) or in grade 4 (‘very effective—as good as no symptoms) of GAE, while this result was obtained only by 32% of controls. The phytosome technology is a 100% food-grade technology applied to natural products in order to obtain a solid dispersion by mixing a carrier such as phospholipids (soy or sunflower lecithin) and the active principles. This formulative approach allows us to increase the exposure area of natural molecules or extracts to the gastrointestinal layer optimizing the relevant biological effects of botanicals.

Another strength of this study is that CASP was well-accepted and tolerated by the participants. Relevant side effects were not reported during the study period.

Some limitations of our study need to be taken into account. To our knowledge, this is the first controlled study on this topic and additional and larger trials need to be performed before achieving a final conclusion. Moreover, studies should be confirmed in order to distinguish subjects with acute infectious diarrhea and subjects with acute non-infectious diarrhea. Another limitation of the study may be the need to identify the bioactive compounds of CASP that could be responsible for the antibacterial activity.

## 5. Conclusions

In conclusion, based on the experience of this trial, we hypothesize that the lecithin-based delivery form of *Boswellia serrata* extract could be a useful and welcome addition to the treatment of acute diarrhea in adults. CASP is safe and reduces the time of becoming healthy, the frequency of stools, the abdominal pain and nausea of subjects with acute diarrhea. Further studies are needed to confirm these promising results.

## Figures and Tables

**Figure 1 nutrients-14-01858-f001:**
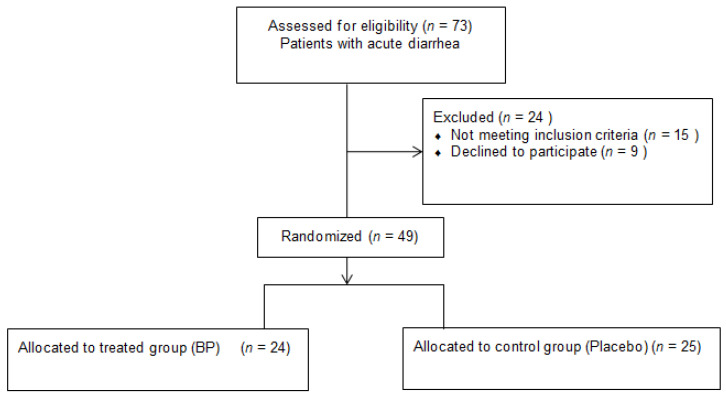
Flow diagram of the participants.

**Figure 2 nutrients-14-01858-f002:**
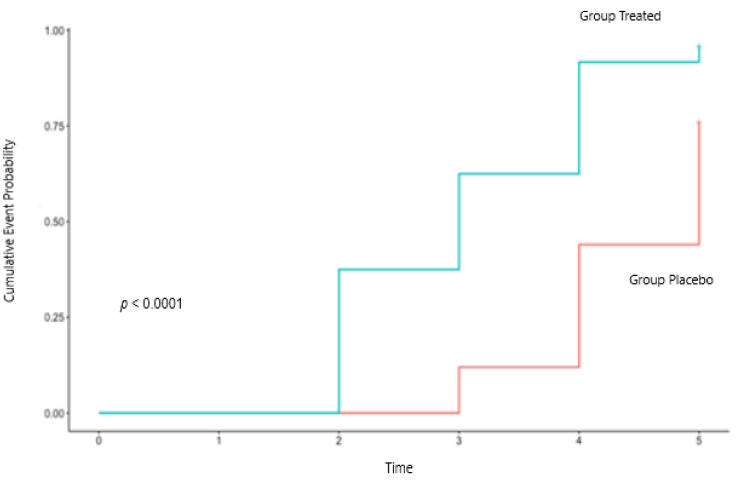
Kaplan-Meier curves in the two groups of the cumulative event probability of stoppage diarrhea. The length of the horizontal lines along the X-axis represents the duration of the time interval, while the vertical lines represent the change in the cumulative event probability.

**Figure 3 nutrients-14-01858-f003:**
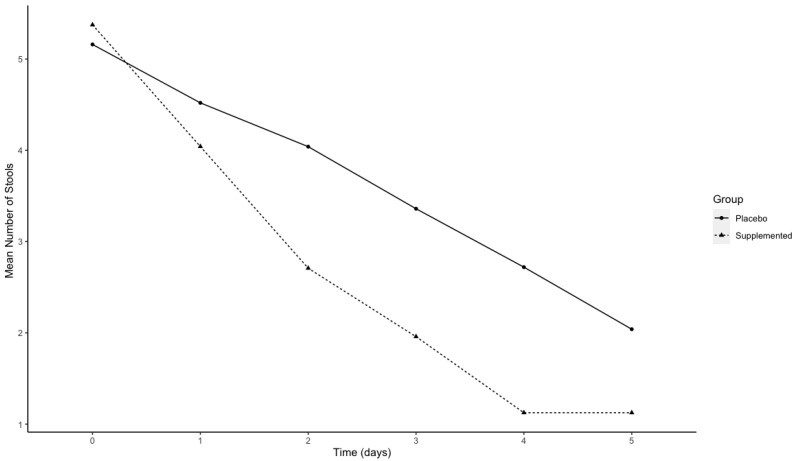
Graphical representation of the mean number of stools over time separately in the two groups: placebo (solid line) and supplemented CASP (dotted line).

**Table 1 nutrients-14-01858-t001:** Baseline participant characteristics.

	Treated Group (*n* = 24) Mean (SD)	Control Group (*n* = 25) Mean (SD)	*p*-Value ^a^
**Age (years)**	50.37 (16.31)	49.84 (14.77)	0.90
**Number of stools at baseline**	5.37 (1.99)	5.16 (1.82)	0.70
	*n* (%)	*n* (%)	
**Sex**			
**Female** **Male**	13 (54.17) 11 (45.83)	14 (56.00) 11 (44.00)	1

^a^*p*-value for between group comparisons.

**Table 2 nutrients-14-01858-t002:** Mean (SD) number of stools and duration of diarrhea in the two groups and *p*-value of the difference between the two groups.

	Supplemented Group (*n* = 24)	Placebo Group (*n* = 25)	*p*-Value
**Mean number of stool reported on day 0**	5.37 (2.00)	5.16 (1.82)	0.85
**Mean number of stool reported on day 1**	4.04 (1.63)	4.52 (1.64)	0.34
**Mean number of stool reported on day 2**	2.71 (1.57)	4.04 (1.37)	0.02
**Mean number of stool reported on day 3**	1.96 (1.12)	3.36 (1.44)	0.002
**Mean number of stool reported on day 4**	1.12 (0.80)	2.72 (1.81)	0.002
**Mean number of stool reported on day 5**	1.12 (0.61)	2.04 (1.79)	0.045
**Duration of diarrhea (days)**	3.08 (1.02)	4.44 (0.71)	<0.0001

**Table 3 nutrients-14-01858-t003:** Frequencies of GAE (global assessment of efficacy), vomiting, nausea, abdominal pain, and abdominal distension at the end of study in the two groups and *p*-value of the difference between the groups.

	Treated Group (*n* = 24) *n* (%)	Placebo Group (*n* = 25) *n* (%)	*p*-Value
**GAE**			
**1** **2** **3** **4**	1 (4.17) 2 (8.33) 10 (41.67) 11 (45.83)	6 (24.00) 11 (44.00) 7 (28.00) 1 (4.00)	0.0002
**Vomiting**			
**0** **1** **2**	23 (95.83) 1 (4.17) 0 (0)	22 (88.00) 2 (8.00) 2 (4.00)	1
**Nausea**			
**0** **1** **2**	19 (79.17) 5 (20.83) 0 (0)	11 (44.00) 10 (40.00) 4 (16.00)	0.02
**Abdominal Pain**			
**0** **3** **6** **10**	16 (66.67) 5 (20.83) 3 (12.50) 0 (0)	8 (32.00) 4 (16.00) 8 (32.00) 5 (20.00)	0.02
**Abdominal Distension**			
**0** **1** **2** **3**	9 (37.5) 8 (33.33) 6 (25.00) 1 (4.17)	7 (28.00) 6 (24.00) 10 (40.00) 2 (8.00)	0.63

## Data Availability

The data presented in this study are available in the text.

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
