# Peer review of "Positive Effects of a Lecithin-Based Delivery Form of Boswellia serrata Extract in Acute Diarrhea of Adult Subjects"

_nutrients, 2022, doi:10.3390/nu14091858_

Round 1
Reviewer 1 Report
the authors of the manuscript "Positive effects of .....adult subjects" is a well written manuscript trying to find a new medication for acute diarrhea in adult subjects.
the authors have used quite a good number of subjects but if there would be more then the statistical analysis would be stronger but its quite good. the authors at the end of the study, 87.5% of the CASP group were classified in effective or very effective grade which is excellent.
the authors did a great job in discussion part with showing different aspect of Boswelia serrata leading to different targets and causing quick recovery from acute diarrhea.
Reviewer 2 Report
Double-blind design is very important and useful result. I would like to know what you use for control group. You should mention about the control, you didn’t give any formula or some placebo capsules?
And you should check the space between words, which sometime double or single.
